

# Approaching the Tsirelson bound with a Sagnac source of polarization-entangled photons

Sandra Meraner, Robert J. Chapman[⋆ †], Stefan Frick, Robert Keil, Maximilian Prilmüller and Gregor Weihs

Institut für Experimentalphysik, Universität Innsbruck, Technikerstraße 25, 6020 Innsbruck, Austria

⋆ robert.chapman@uibk.ac.at

† Current address: CQC2T, School of Engineering, RMIT University, Melbourne, Australia

## Abstract

High-fidelity polarization-entangled photons are a powerful resource for quantum communication, distributing entanglement and quantum teleportation. The Bell-CHSH inequality $S \leq 2$ is violated by bipartite entanglement and only maximally entangled states can achieve $S = 2\sqrt{2}$, the Tsirelson bound. Spontaneous parametric down-conversion sources can produce entangled photons with correlations close to the Tsirelson bound. Sagnac configurations offer intrinsic stability, compact footprint and high collection efficiency, however, there is often a trade off between source brightness and entanglement visibility. Here, we present a Sagnac polarization-entangled source with $2\sqrt{2} - S = (5.65 \pm 0.57) \times 10^{-3}$, on-par with the highest $S$ parameters recorded, while generating and detecting $(4660 \pm 70)\,\mathrm{pairs/s/mW}$, which is a substantially higher brightness than previously reported for Sagnac sources and around two orders of magnitude brighter than for traditional cone sources with the highest $S$ parameters. Our source records $0.9953 \pm 0.0003$ concurrence and $0.99743 \pm 0.00014$ fidelity to an ideal Bell state. By studying systematic errors in Sagnac sources, we identify that the precision of the collection focal point inside the crystal plays the largest role in reducing the $S$ parameter in our experiment. We provide a pathway that could enable the highest $S$ parameter recorded with a Sagnac source to-date while maintaining very high brightness.



# 1  Introduction

Polarization-entangled photons have demonstrated striking quantum phenomena such as quantum teleportation [1, 2], multi-photon entanglement [3], long-distance quantum communication [4] and loophole-free Bell tests [5, 6]. Traditional cone (non-colinear) spontaneous parametric down conversion (SPDC) sources have been the workhorse of quantum photonics experiments for the past decades [7]. However, their geometry limits the photon flux as the majority of generated pairs are discarded and they have large footprints to spatially separate the pump laser from the converted photons. Sagnac interferometer sources occupy minimal space and utilize colinear SPDC in periodically poled crystals, meaning no generated photons are rejected [8–15]. They also enable very high fidelity Bell state generation as only the propagation directions must be indistinguishable which can be straightforward to implement unlike, for example, spectral indistinguishability.

The Bell-CHSH (Clauser, Horne, Shimony and Holt) experiment is a standard for entanglement verification [16]. Violating the Bell-CHSH inequality $S \leq 2$ certifies experimental results that cannot be reconciled with any classical model of reality. The Tsirelson bound, at $S = 2\sqrt{2}$, is the upper limit of $S$ that any bipartite entangled state can achieve [17]. Also certain fundamental restrictions on the information content of quantum states can be associated with this bound [18–20]. Developing photon-pair sources at the Tsirelson bound could, thus, help explore these principles at the limits of quantum theory and, furthermore, have applications in quantum computing protocols such as teleportation [2]. Previous non-colinear SPDC experiments have sought to reach the Tsirelson bound [21, 22] with the lowest value of $2\sqrt{2} - S$ reported as $(8.4 \pm 5.1) \times 10^{-4}$, with a source brightness of $\approx 63$ pairs/s per mW of pump power [23]. Sagnac sources can achieve substantially higher brightness due to the colinear pair generation. The closest to the Tsirelson bound a Sagnac source has achieved is $2\sqrt{2} - S = (3.13 \pm 3.50) \times 10^{-3}$ with a brightness of $\approx 700$ pairs/s/mW [10].

Here, we present a Sagnac interferometer source of polarization-entangled photons that optimizes both the Bell-CHSH violation and brightness. We record $2\sqrt{2} - S = (5.65 \pm 0.57) \times 10^{-3}$ with $(4660 \pm 70)$ pairs/s/mW, that is, at considerably higher brightness than previous experiments. Our source produces maximally entangled Bell $|\Psi^-\rangle$-states with fidelity $F = 0.99743 \pm 0.00014$ and concurrence $C = 0.9953 \pm 0.0003$ without the need of accidental subtraction. We thoroughly study systematic errors, statistical uncertainties and post-processing to investigate the impact on the $S$ parameter. We identify that the position of the collection focus inside the crystal and the balancing of pump power in the interferometer

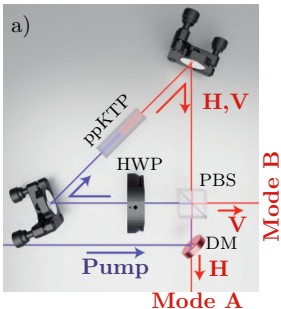
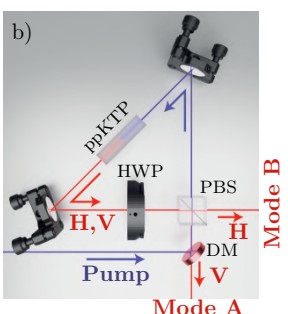
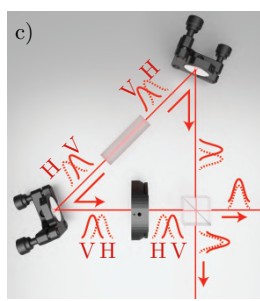

Figure 1: A Type-II SPDC source in a Sagnac interferometer. The a) clockwise and b) counter-clockwise direction laser pump generates orthogonally polarized photon-pairs that are separated at the polarizing beam splitter (PBS). The pump is rejected with the dichroic mirror (DM). The half-wave plate (HWP) swaps the polarization of the clockwise pump and counter-clockwise photon-pair. c) The ppKTP birefringence causes a temporal walk-off between the horizontally and vertically polarized photons. A Bell state can only be prepared if the temporal walk-off is equal for clockwise and counter-clockwise propagation.

are the dominant factors limiting our source and we predict that, with feasible improvements, our source could halve the gap to the Tsirelson bound without reducing the high brightness.

## 2 Sagnac source of polarization-entangled photons

We generate polarization-entangled photon-pairs with a periodically poled potassium titanyl phosphate (ppKTP) crystal designed for Type-II SPDC. Figure 1 shows clockwise ($\circlearrowright$) and counter-clockwise ($\circlearrowleft$) propagation in the Sagnac interferometer. The $\circlearrowright$ ($\circlearrowleft$) propagating pump laser produces a horizontally (vertically) polarized photon in output mode $A$ and a vertically (horizontally) polarized photon in output mode $B$. By generating a single photon-pair and erasing the "which direction" information, we prepare the entangled Bell state $|\Psi^-\rangle = \frac{1}{\sqrt{2}}(|H_A V_B\rangle - |V_A H_B\rangle)$. The Sagnac interferometer has intrinsic stability as both directions have the same optical path, however, to achieve the maximum fidelity Bell state, the crystal must be centered at the focus point of the collection optics. This ensures the birefringent walk-off values experienced by both propagation directions are equal and can be compensated by reversing the polarization of the $\circlearrowleft$ direction using a half-wave plate (HWP), as illustrated in Fig. 1c.

The schematic of our experiment is presented in Fig. 2. We pump the Sagnac source with a 403.9 nm wavelength continuous-wave laser and control the power with HWP H1, polarizing beam splitter (PBS) PBS1 and a beam block. We set the polarization of the laser to diagonal with quarter-wave plate (QWP) Q1 and HWP H2 which ensures equal power traveling $\circlearrowright$ and $\circlearrowleft$ in the Sagnac interferometer. The laser is focused at the center of the temperature-controlled ppKTP crystal to generate 807.8 nm wavelength photons and we suppress the pump laser with a dichroic mirror, colored glass and interference filters. We perform projection measurements on each photon using a QWP, HWP and polarizer. We record the photon arrival times with efficient (>60 %) superconducting single photon detectors and a 3 ps resolution time correlator. Details of temperature tuning the ppKTP crystal and characterization of the wave plates are in Appendix A and Appendix B.

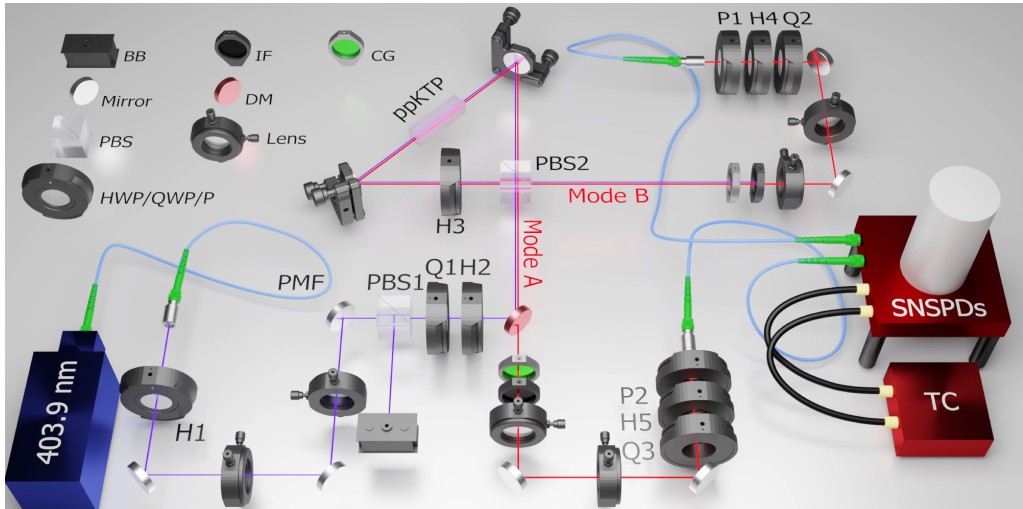

Figure 2: Schematic of the polarization-entangled Sagnac source. We generate orthogonally polarized photon-pairs at 807.8 nm wavelength with a ppKTP crystal in a Sagnac interferometer and, by erasing the "which path" information of the pump with PBS2, we prepare nearly ideal Bell $|\Psi^-\rangle$-states. PMF, Polarization maintaining fiber; BB, Beam block; IF, interference filter; CG, colored glass; HWP, Half-wave plate; QWP, Quarter-wave plate; P, Polarizer; DM, Dichroic mirror; PBS, Polarizing beam splitter; SNSPDs, Superconducting nanowire single photon detectors; TC, time correlator.

## 3   Bell-CHSH inequality violation

While the violation of the Bell-CHSH inequality is routinely performed in quantum optics laboratories [24–30], measuring the Tsirelson bound with maximally entangled photons is a greater challenge that requires high fidelity state preparation, low statistical noise and precise measurement control. Nevertheless, achieving very high values for $S$ is important for testing the foundations of quantum mechanics as a violation of the Tsirelson bound would require new theoretical frameworks and invalidate quantum mechanics. Under the fair-sampling assumption, the $S$ parameter is calculated from four expectation values, $S = E_0 + E_1 - E_2 + E_3$, with

$$E_i = \frac{n(\alpha_i, \beta_i) - n(\alpha_i + \frac{\pi}{2}, \beta_i) - n(\alpha_i, \beta_i + \frac{\pi}{2}) + n(\alpha_i + \frac{\pi}{2}, \beta_i + \frac{\pi}{2})}{n(\alpha_i, \beta_i) + n(\alpha_i + \frac{\pi}{2}, \beta_i) + n(\alpha_i, \beta_i + \frac{\pi}{2}) + n(\alpha_i + \frac{\pi}{2}, \beta_i + \frac{\pi}{2})}, \tag{1}$$

$$n(\alpha_i, \beta_i) = N\tau \langle R(\alpha_i)_A R(\beta_i)_B | \rho | R(\alpha_i)_A R(\beta_i)_B \rangle, \tag{2}$$

where $\rho$ is the two-qubit entangled state, $N$ is the total coincidence rate and $\tau$ is the integration time. $n(\alpha_i, \beta_i)$ is the number of coincidence events recorded with the polarizer on output mode $A$ ($B$) at angle $\alpha_i$ ($\beta_i$). This corresponds to projecting mode $j \in \{A, B\}$ onto the state $|R(\alpha_i)_j\rangle = \cos(\alpha_i)|H_j\rangle + \sin(\alpha_i)|V_j\rangle$. The measurement angles that give the maximum $S$ parameter depend on the quantum state and for a $|\Psi^-\rangle$ state, we use $\alpha = \{0, \frac{\pi}{4}, 0, \frac{\pi}{4}\}$ and $\beta = \{\frac{\pi}{8}, \frac{\pi}{8}, \frac{3\pi}{8}, \frac{3\pi}{8}\}$. A bipartite quantum state with $S > 2$ cannot be described by local-realistic theories, even if supplemented by local hidden variables, and the Tsirelson bound, with $S = 2\sqrt{2}$, can only be achieved with maximally entangled states.

We repeat the Bell-CHSH experiment 25 times, performing the 16 projections using motor-controlled wave plates and fixed polarizers. We use $\tau = 60\,\text{s}$ integration time per measurement setting and record a coincidence rate of $N = (4100 \pm 70)$ pairs/s at 0.88 mW pump power. Over the 25 repetitions, we record a total of 24 602 439 coincidence events and estimate the number

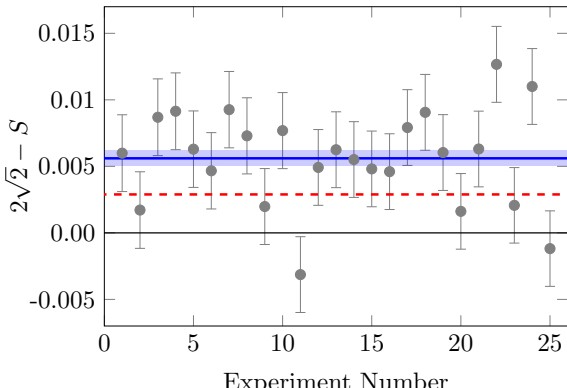

Figure 3: The $S$ parameter measured for 25 repetitions of the Bell-CHSH experiment with Poissonian uncertainty in the photon count statistics. The horizontal blue line is the combined result and shaded region is the one sigma uncertainty bound. The red dashed line is the predicted value with a further optimized setup.

of photon pairs before projection to be $\approx 10^8$. We achieve high brightness by constructing a well-aligned colinear photon-pair source, where all generated photons can be utilized, and by optimizing the output collection into single mode optical fibers. Cone sources suffer from low brightness as most of the photons are discarded, meaning far higher pump powers are required for the same fiber-coupled photon-pair flux. We measure around two orders of magnitude higher brightness than the cone source with the record high $S$ parameter [23]. We present the results of each Bell-CHSH experiment in Fig. 3 and, by summing the coincidence counts from all trials, we calculate a final value of $2\sqrt{2} - S = (5.65 \pm 0.57) \times 10^{-3}$. The uncertainty here assumes Poissonian counting statistics, where each measurement uncertainty is $\sqrt{n(\alpha_i, \beta_i)}$. We perform uncertainty propagation with Eq. 2 to calculate the uncertainty in the measured $S$ parameter (see Appendix C for detailed results and the full uncertainty calculations). Figure 3 shows that some trials record $S > 2\sqrt{2}$ which we attribute to Poissonian fluctuations in photon count statistics.

We perform quantum state tomography (QST) before and after the Bell-CHSH experiment to characterize the two-qubit state we prepare. QST on two qubits requires a minimum of 16 projection measurements and solving a linear inversion problem with the recorded coincidence events [31]. While maximum likelihood estimation (MLE) has been shown to have drawbacks [32], we find it necessary to recover a physical state with positive eigenvalues and trace one. We present the density matrix recorded after the Bell-CHSH experiment in Fig. 4a, which has a fidelity to the state before the experiment of $0.9993 \pm 0.0003$, demonstrating that our setup is stable over several hours. The concurrence of the recorded density matrix is $0.9953 \pm 0.0003$ and fidelity to the ideal Bell $|\Psi^-\rangle$ state is $0.99743 \pm 0.00014$. Here, we use Poissonian statistical uncertainty in a Monte-Carlo simulation to calculate the uncertainty bounds and present the distribution of the concurrence and fidelity in Fig. 4b. MLE has been shown to underestimate state fidelity, however, as we operate with near ideal Bell states and with high count rates, this effect is negligible [33]. We explore the impact of MLE in Appendix D.

## 4 Origins of error in Sagnac photon-pair sources

In a Sagnac interferometer source, the probability of generating a photon-pair from the ↻ and ↺ directions must be equal to prepare a Bell state. This probability encompasses both the pair production rate, which is proportional to the laser power, and the coupling efficiency at the

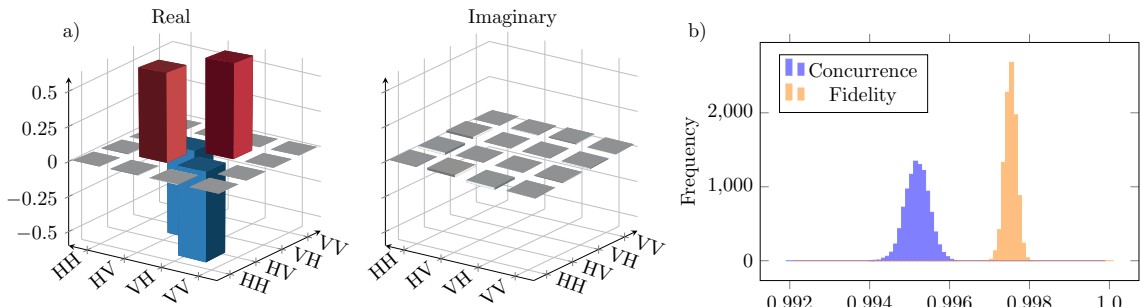

Figure 4: a) The density matrix recorded after the Bell-CHSH experiment. There is a clear offset in the magnitude of the $|HV\rangle$ and $|VH\rangle$ components, otherwise the state is close to the ideal Bell state. b) The fidelity and concurrence distribution calculated from a Monte-Carlo simulation with $10^4$ repetitions and assuming Poissonian counting statistics.

output. The ↻ and ↺ laser power is controlled by Q1 and H2 in the setup and output fiber coupling is controlled using precision mirror mounts. We consider these factors as a single term $P$, where $\frac{P}{2}$ of the pairs are generated by the ↺ propagating pump laser and $1 - \frac{P}{2}$ of the pairs are generated by the ↻ propagating pump laser. This gives the generated state as

$$\sqrt{1 - \frac{P}{2}}\,|HV\rangle - \sqrt{\frac{P}{2}}e^{i\phi}\,|VH\rangle, \tag{3}$$

and $\phi$ is the relative phase that is controlled by the pump polarization. We simulate the impact of varying $P$ on the fidelity, concurrence and $S$ parameter in Fig. 5a and from the imbalance of $|HV\rangle$ and $|VH\rangle$ in our density matrix, we estimate $P = 1.03$ in our experiment. This corresponds to a reduction of the $S$ parameter by $6.4 \times 10^{-4}$. Improving the balance of the Sagnac source to reach $P = 1.01$ would reduce this to $7.1 \times 10^{-5}$.

The Sagnac source is inherently phase stable for the ↻ and ↺ propagation directions, however, the focal point of the collection optics must be at the center of the nonlinear crystal. This ensures the ↻ and ↺ collected photons propagate through equal lengths of the birefringent ppKTP crystal. As shown in Fig. 1c, the HWP swaps the polarization of the ↺ propagating photons such that a coherent state is generated. A longitudinal offset of the crystal position causes an asymmetric change to the temporal distributions for ↻ and ↺ down-converted photons. This leads to distiguishability of the ↻ and ↺ generated photons at the PBS and reduced visibility. We simulate this offset by convolving the temporal wave-packet of the generated single photons

$$I_p(t) = \frac{\Delta\omega}{\sqrt{2\pi}}\, e^{\frac{-t^2}{2}\Delta\omega^2}, \tag{4}$$

with the photon-pair collection probability for different generation positions inside the crystal. Whereas the generation probability is uniform over the crystal length, the collection probability is not uniform but depends on the geometries of the pump and collection modes [34]. The full-width at half-maximum (FWHM) duration of the wavepackets $\frac{2\sqrt{2\ln 2}}{\Delta\omega} \approx 1.92$ ps is inferred from the measured $\approx 0.5$ nm FWHM spectrum of our down-converted photons. The photon-pair collection probability is given by the magnitude square of the spatial overlap ($\mathcal{O}_s$) of the signal ($s$), idler ($i$) and pump ($p$) fields

$$\mathcal{O}_s \propto w_p w_s w_i (q_s^* q_i^* + q_p q_i^* + q_p q_s^*)^{-1}, \tag{5}$$

$$q_j = w_j^2 + \frac{2i(z - z_{0,j})}{k_j}, \tag{6}$$

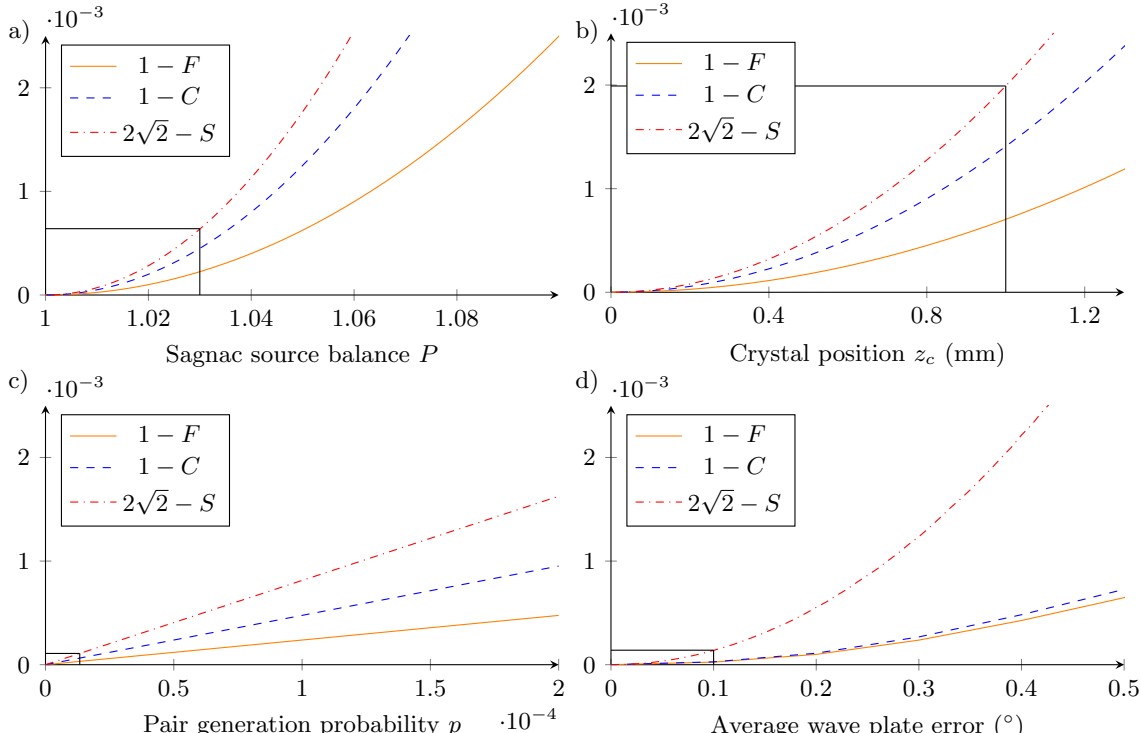

Figure 5: The main error sources that reduce the $S$ parameter in a Sagnac source. Simulation results for the fidelity, concurrence and $S$ parameter for varying a) Sagnac source balance, b) crystal position offset, c) accidental multi-pair generation and d) projection measurement wave plate errors. Black lines indicate the results from our experiment. Our result in a) is extracted from the density matrix and in c) from the measured coincidence-to-single ratio. In b) and d), our result is estimated from our hardware.

where $w_j$ is the waist size, $z_{0,j}$ is the collection focus position, $z$ is the position inside the crystal and $k_j$ is the wavenumber for field $j$ [34]. Considering the dispersion and birefringence of the ppKTP crystal [35–37], we calculate the probability distribution of photon-pair collection for the $\circlearrowright$- and $\circlearrowleft$- propagating pump laser and different crystal positions. After translating formula (5) into time coordinates and convolving with (4), we can calculate probability densities of generation times for different focal points and for both axes of the birefringent crystal. These probability distributions between $\circlearrowright$ and $\circlearrowleft$ directions must coincide with high overlap $\mathcal{O}_c$ at the PBS to generate a high-fidelity entangled state. We simulate the generated quantum state with a crystal position of $z_c$ as

$$\rho(z_c) = \frac{1}{2}(\mathcal{O}_c^o + \mathcal{O}_c^e)\,|\Psi^-\rangle\langle\Psi^-| + \tag{7}$$
$$\frac{2 - \mathcal{O}_c^o - \mathcal{O}_c^e}{4}(|HV\rangle\langle HV| + |VH\rangle\langle VH|)$$

where $\mathcal{O}_c^o$ ($\mathcal{O}_c^e$) is the temporal overlap (normalized to 1 over the crystal length) of the ordinary (extraordinary) polarization. The complete derivation of $\rho(z_c)$ is in Appendix E. In Fig. 5b. we plot the fidelity, concurrence and $S$ parameter against the offset in the focal position. We estimate 1.0 mm accuracy of our ppKTP crystal position from the center of the Sagnac interferometer by the precision of the ruler used to measure it. An error of 1.0 mm corresponds to a $2.0 \times 10^{-3}$ reduction in the $S$ parameter. This is a significant decrease and a key reason our $S$ parameter is lower than the Tsirelson bound. We also consider offsets in the positions

Table 1: Summary of error sources in a Sagnac interferometer that degrade the $S$ parameter, listed in order of descending impact.

| Error source | Value | Reduced $S$ parameter |
|---|---|---|
| Crystal position ($z_c$) | 1.0 mm | $2.0 \times 10^{-3}$ |
| Sagnac source balance ($P$) | 1.03 | $6.4 \times 10^{-4}$ |
| Wave plate zero-point and retardance | See Appendix B | $1.9 \times 10^{-4}$ |
| Wave plate setting error | $\pm 0.1°$ | $1.4 \times 10^{-4}$ |
| Multi-pair generation ($p$) | $1.3 \times 10^{-5}$ | $1.1 \times 10^{-4}$ |
| Total | | $3.1 \times 10^{-3}$ |
| Experiment | | $(5.65 \pm 0.57) \times 10^{-3}$ |

of the focal lenses for the $s$, $i$ and $p$ fields, however, because mode $A$ ($B$) always collects $s$ ($i$) photons of both generation directions, any offset is compensated due to the symmetry of the Sagnac interferometer.

SPDC sources must operate with low pair-generation rates to suppress parasitic multi-pair emission that degrades the photon-pair state purity. In a Sagnac source, multi-photon events can occur in a single direction, either double ↻ or ↺ down-conversion, or from the simultaneous creation of both a ↻ and a ↺ pair. The rate of such events can be estimated from the measured rates of singles and coincidences. We obtain a ratio of double to single-pair emissions of $p = 1.3 \times 10^{-5}$, which reduces $S$ by $1.1 \times 10^{-4}$ (see Appendix F). In Fig. 5c we plot the impact of multi-pair emission on the fidelity, concurrence and $S$ parameter. Hence, at the employed pump power the contribution of multi-pairs to the systematic errors is small compared to the other two effects discussed above.

QST and the Bell-CHSH experiment rely on precise wave plate and polarizer settings to perform the necessary projection measurements. We use stepper motor controlled wave plate rotators with $\pm 0.1°$ repeatability specified by the manufacturer and we approach each angle from the same direction to avoid backlash errors. We perform a Monte-Carlo simulation of both the Bell-CHSH experiment and QST with wave plate precision as the only source of error. In Fig. 5d we present the reduced fidelity, concurrence and $S$ parameter with wave plate errors up to 0.5°. For a wave plate uncertainty of $\pm 0.1°$, the average $S$ parameter is reduced by $1.4 \times 10^{-4}$. In our experiment, we can see that wave plate precision is not a major factor reducing the $S$ parameter. We characterize each wave plate using a polarimeter to find the exact retardance and zero-point angle (see Appendix B). From the measured polarization rotations, we have imperfect retardance of each waveplate as well as uncertainty from the fit. We therefore run an additional Monte-Carlo simulation taking into account the measured wave plate retardances and the uncertainty in the retardance and zero-point angles from which we extract an additional $1.9 \times 10^{-4}$ reduction in $S$. Wave plate rotations also cause beam steering that can lead to projection measurement-dependent loss. By recording the single count rates (not coincidences) for different wave plate angles, we can estimate the projection-dependent loss and extract an estimated reduction in the $S$ parameter. For one channel we estimate a 5% modulation which corresponds to a reduction in $S$ of $2.1 \times 10^{-5}$ and the other channel has near equal coupling for all wave plate angles. We therefore consider this a very minor impact on the measured Bell-CHSH violation.

## 5 Conclusion

Table 1 summarizes these sources of error and the impact they have on the $S$ parameter. We have identified that the position of the crystal in the Sagnac loop and the balance of the two

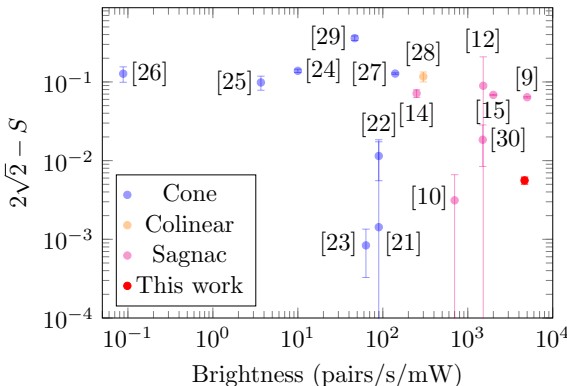

Figure 6: Reported $S$ parameters against the source brightness.

emission directions are the largest known factors reducing the $S$ parameter in our experiment. There is still a gap between the experimentally recorded value and the predicted value of $2.55 \times 10^{-3}$ that has not been accounted for. A 1.5 mm offset of the focal positions inside the crystal would account fully for this gap, however, at this point we cannot rule out further error sources. For example, this gap could originate from non-overlapping collection points leading to distinguishability of the generated photons, or from dispersion caused by the HWP inside the Sagnac interferometer that only affects the ↻ propagating photon pair. Figure 6 compares this work with published Bell-CHSH SPDC experiments in terms of source brightness and the measured $S$ parameter.

The error sources identified here can be readily improved by controlling the pump laser polarization, optimizing the fiber coupling and focal position and improving the wave plate characterization and motor precision. We could practically improve the Sagnac source balance to $P = 1.01$, the focal-point precision to 0.1 mm and the wave plate error to 0.01°. With these improvements and ideal wave plates, we predict an $S$ parameter of $2\sqrt{2} - S = 2.64 \times 10^{-3}$ which includes the unknown errors in our experiment. This would be the highest $S$ parameter reported for a Sagnac source while maintaining the measured high brightness of 4660 pairs/s/mW that reduces the acquisition time and statistical uncertainty.

**Funding information** The authors acknowledge funding by the Austrian Science Fund (grants W 1259, I 2562, P 30459 and F 71) and the European Union's Horizon 2020 research and innovation programme (grant agreement No 820474).

# Appendices

## A Crystal temperature

The ppKTP crystal has a poling period of 9.825 μm and a length of 15 mm and is temperature controlled. We heat the crystal and measure the spectrum of the signal and idler photons. In Fig. A1 we plot the center wavelength against temperature and find the degenerate point at $(31.9 \pm 0.3)$ °C.

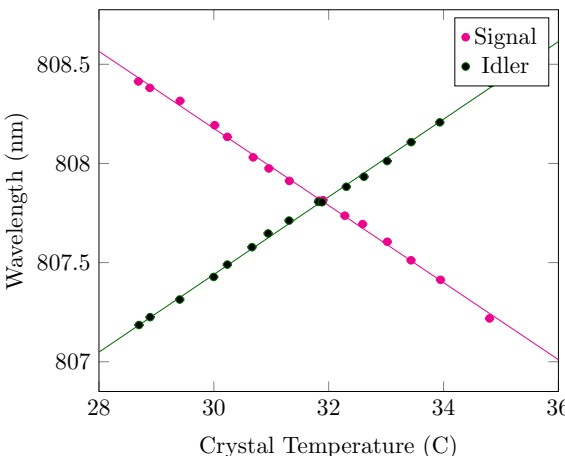

Figure A1: Signal and idler wavelengths vs crystal temperature.

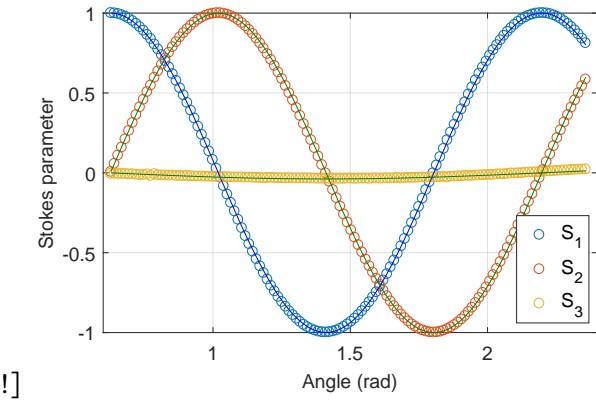

][h!]

Figure A2: Stokes parameters for one of the half wave plates in our setup.

Table A1: Summary of the waveplate characterization.

| Wave plate | Retardance (rad) | Retardance (deg) | Zero point (rad) | Zero point (deg) |
|---|---|---|---|---|
| HWP 1 | $(1.0122 \pm 0.0035)\pi$ | $182.196 \pm 0.63$ | $0.627 \pm 0.00010$ | $35.924 \pm 0.0057$ |
| QWP 1 | $(1.0427 \pm 0.0005)\pi/2$ | $93.843 \pm 0.045$ | $0.602 \pm 0.00024$ | $34.492 \pm 0.0138$ |
| HWP 2 | $(1.0075 \pm 0.0030)\pi$ | $181.35 \pm 0.54$ | $0.454 \pm 0.00015$ | $26.012 \pm 0.0086$ |
| QWP 2 | $(0.99155 \pm 0.0005)\pi/2$ | $89.2395 \pm 0.045$ | $1.918 \pm 0.00018$ | $109.893 \pm 0.0103$ |

# B  Wave plate characterization

The four analysis wave plates were individually characterized using a polarimeter (Thorlabs PAX5720IR2-T). The Stokes parameters are measured, from which we calculate the wave plate retardance and the zero point offset which is caused predominantly be the mounting position. Figure A2 presents the characterization of the half wave plate in the signal arm. Table A1 presents the results for the four wave plates. We use these values to optimize the projection measurements in our CHSH experiments.

# C   Bell-CHSH measurement results and error propagation

The complete measurement results for our CHSH experiment are presented in table A2. The QWPs are also rotated to prepare the ideal projection measurements for the CHSH experiment. We perform 25 repetitions of the 16 projection measurements and sum the results. For each projection measurement we integrate the coincidence rate for 60 seconds. The wave plate angles include compensation for manufacturing imprecision of the optical axis position.

Table A2: Bell-CHSH experimental results

| Mode A HWP (deg) | Mode A QWP (deg) | Mode B HWP (deg) | Mode B QWP (deg) | Coincidence Count |
|---|---|---|---|---|
| 25.9 | 19.5 | 47.0 | 146.4 | 431 677 |
| 25.9 | 19.5 | 70.7 | 103.3 | 2 686 277 |
| 25.9 | 19.5 | 90.3 | 148.3 | 2 582 292 |
| 25.9 | 19.5 | 114.8 | 101.7 | 445 574 |
| 71.5 | 20.3 | 47.0 | 146.4 | 2 567 446 |
| 71.5 | 20.3 | 70.7 | 103.3 | 460 054 |
| 71.5 | 20.3 | 90.3 | 148.3 | 429 585 |
| 71.5 | 20.3 | 114.8 | 101.7 | 2 671 279 |
| 48.2 | 63.9 | 47.0 | 146.4 | 501 199 |
| 48.2 | 63.9 | 70.7 | 103.3 | 2 659 125 |
| 48.2 | 63.9 | 90.3 | 148.3 | 427 316 |
| 48.2 | 63.9 | 114.8 | 101.7 | 2 613 659 |
| 93.8 | 64.8 | 47.0 | 146.4 | 2 575 410 |
| 93.8 | 64.8 | 70.7 | 103.3 | 440 230 |
| 93.8 | 64.8 | 90.3 | 148.3 | 2 626 399 |
| 93.8 | 64.8 | 114.8 | 101.7 | 484 917 |
| | | | Total | 24 602 439 |

In the CHSH experiment, there are four measurement settings $i = 0, \ldots, 3$ and, for each measurement setting, four projection measurements are required for the angles $(\alpha_i, \beta_i)$, $(\alpha_i, \beta_i + \frac{\pi}{2})$, $(\alpha_i + \frac{\pi}{2}, \beta_i)$ and $(\alpha_i + \frac{\pi}{2}, \beta_i + \frac{\pi}{2})$. For brevity we label the measurement results as $n(\alpha_i, \beta_i) = n_{i,++}$, $n(\alpha_i, \beta_i + \frac{\pi}{2}) = n_{i,+-}$, etc.. The main source of uncertainty in our experiment is from Poissonian counting statistics. Here, the uncertainty on the measurement $n_{i,++}$ is $\Delta n_{i,++} = \sqrt{n_{i,++}}$ and we assume the counts for all measurement uncertainties are independent.

The CHSH inequality measures

$$S = E_0 + E_1 - E_2 + E_3, \tag{A1}$$

$$E_i = \frac{n_{i,++} - n_{i,+-} - n_{i,-+} + n_{i,--}}{n_{i,++} + n_{i,+-} + n_{i,-+} + n_{i,--}} = \frac{N_i}{D_i}. \tag{A2}$$

The uncertainty in $S$ can be calculated as

$$\Delta S = \sqrt{\Delta E_0^2 + \Delta E_1^2 + \Delta E_2^2 + \Delta E_3^2}, \tag{A3}$$

$$\Delta E_i = \sqrt{\left(\frac{\partial E_i}{\partial n_{i,++}}\right)^2 \Delta n_{i,++}^2 + \left(\frac{\partial E_i}{\partial n_{i,+-}}\right)^2 \Delta n_{i,+-}^2 + \left(\frac{\partial E_i}{\partial n_{i,-+}}\right)^2 \Delta n_{i,-+}^2 + \left(\frac{\partial E_i}{\partial n_{i,--}}\right)^2 \Delta n_{i,--}^2}, \tag{A4}$$

$$\Delta n_{i,ab} = \sqrt{n_{i,ab}}. \tag{A5}$$

Finally by the quotient rule, we can calculate the derivatives of $E_i$ as

$$\frac{\partial E_i}{\partial n_{i,++}} = \frac{\partial E_i}{\partial n_{i,--}} = \frac{1}{D_i} - \frac{N_i}{D_i^2} \tag{A6}$$

$$\frac{\partial E_i}{\partial n_{i,+-}} = \frac{\partial E_i}{\partial n_{i,-+}} = -\frac{1}{D_i} - \frac{N_i}{D_i^2}, \tag{A7}$$

such that one obtains with eqs. (A4) and (A5):

$$\Delta E_i = \frac{2}{D_i^{3/2}} \sqrt{(n_{i,++} + n_{i,--})(n_{i,+-} + n_{i,-+})}. \tag{A8}$$

# D   Maximum likelihood estimation

We perform a quantum state tomography (QST) Monte Carlo simulation with different photon count rates. We assume here that the uncertainty in the photon count rate is given by the Poissonian counting uncertainty alone. It has been demonstrated that maximum likelihood estimation (MLE) sometimes underestimate the state fidelity [33]. Figure A3 presents the state fidelity, concurrence and $S$ parameter for an ideal Bell state after Monte-Carlo simulation of QST and MLE. It is clear that, with $> 10^4$ counts per measurement, the impact is negligible. In our experiment, we have $> 10^6$ coincidence counts per measurement setting and can therefore neglect MLE as a source of error.

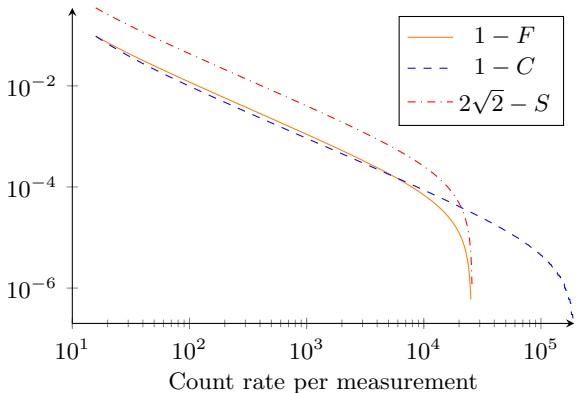

Figure A3: Fidelity, concurrence and $S$ parameter for a Monte Carlo simulation of QST with MLE on an ideal Bell state. With high count rates $> 10^4$, QST has negligible difference from the ideal state.

QST using maximum likelihood estimation (MLE) is also known to give rise to inaccurate state reconstruction [32]. The results in the main text do employ MLE as it is necessary to recover physical density matrices. Here, for verification we also perform linear QST without MLE on the data recorded after the CHSH experiment and plot the resulting density matrix in Fig. A4.

This tomography leads to an unphysical state with small negative eigenvalues and with fidelity >1 to the ideal $|\Psi^-\rangle$ state but with lower concurrence, $0.910 \pm 0.005$. This could suggest that MLE might slightly overestimate the concurrence, especially when measuring states close to the Tsirelson bound. However, from simulations according to the model of our experiment, we estimate our state should achieve $C = 0.997$.

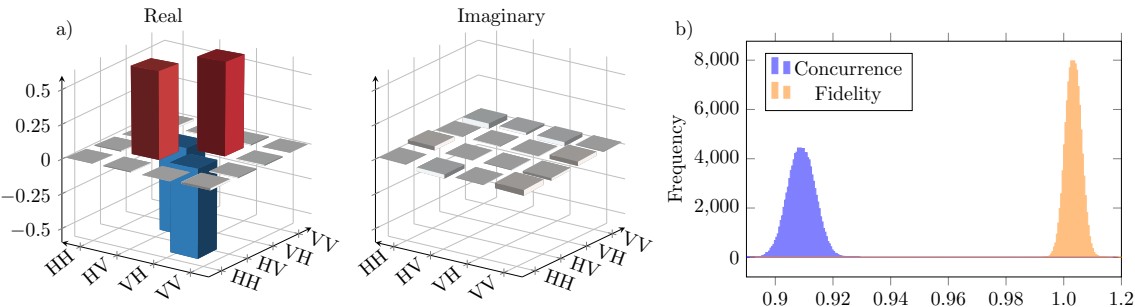

Figure A4: a) Recovered density matrix without using MLE. b) Concurrence from a Monte Carlo simulation and fidelity to the ideal Bell state. It is clear that MLE is necessary to produce a physical density matrix.

# E  Crystal offset

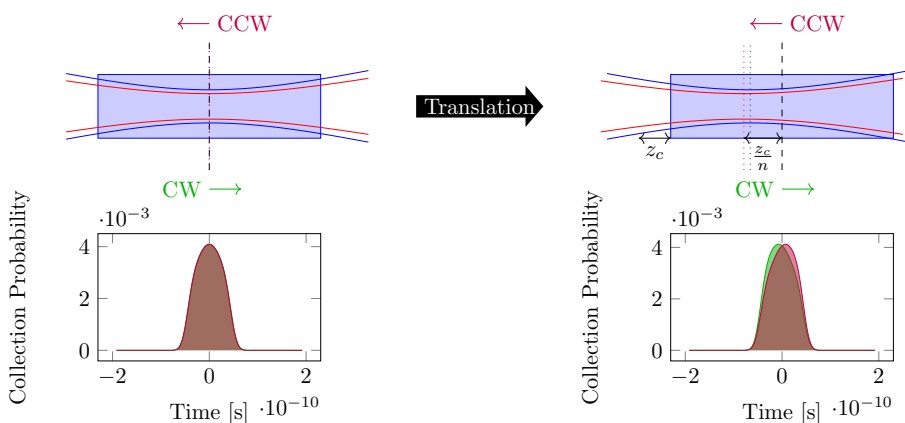

Figure A5: Explanation of the effect of an offset crystal. The left side shows the perfect situation where the crystal is centered in the Sagnac loop and both pump (blue) and collection (of signal or idler photons, red) foci are in the middle of the crystal. Because the collection focus is in the center of the crystal the temporal density of states for ↻ and ↺ propagations perfectly overlap. The right side shows the situation when the crystal is shifted by $z_c$ to the right. A shift of $z_c$ to the right will shift the focal position by $\frac{z_c}{n}$ to the left inside the crystal. This focal offset creates mismatch of collection probability between ↻ and ↺ created photons. Because ↺ created photons are more likely to travel a shorter distance throught the crystal they are more likely to arrive at the PBS earlier. This gives rise to a distinguishability between ↻ and ↺ photons and diminishes state fidelity, concurrence and S parameter.

It is necessary that the crystal is in the center of the Sagnac interferometer, such that the ↻ and ↺ direction SPDC photons experience the same walk-off for each crystal axis which is then compensated with the HWP. If the crystal is not in the center, then the overlap of the $|HV\rangle$ ↻ component and $|VH\rangle$ ↺ component is reduced. The effect of this overlap depends on the coherence length of the photons as well as the geometry of collection and pump beams because photon pairs are more likely to be collected at points with higher spatial overlap between the pump and each collection beam (see fig. A5).

We firstly measure the spectrum of the down-converted photons with a single photon spectrometer (Princeton Instruments) and record a full-width at half-maximum (FWHM) of ≈0.5 nm, corresponding to a temporal FWHM of ≈1.92 ps. The spectrum is plotted in Fig. A6

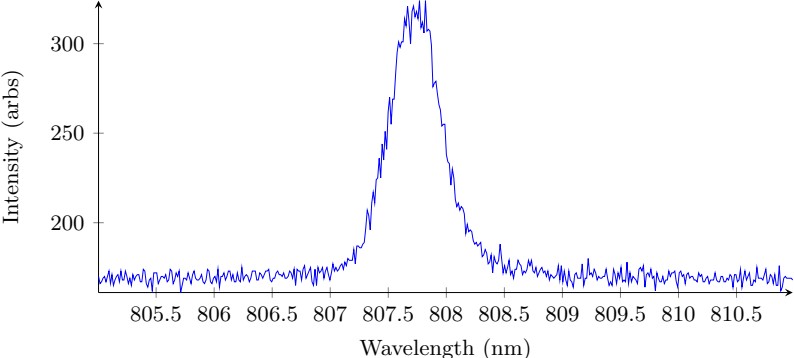

Figure A6: Spectrum of the down-converted photons.

and is Gaussian (instead of a sinc-shape) because of the collection optics in our compact setup are collecting non-flat wave-fronts [34]. We then convolve the temporal wavepacket of a single photon

$$I_p(t) = \frac{\Delta\omega}{\sqrt{2\pi}} e^{\frac{-t^2}{2}\Delta\omega^2},$$ (A9)

with the temporal overlap distribution $\mathcal{O}_s^j(t) = \mathcal{O}_t(v_g^j t)$ with $j \in \{s, i, p\}$ (see Eq. 5 of the main text). The probability density of collection times is obtained by calculating the overlap $\mathcal{O}_t$ depending on the waists and focal position of the pump, signal and idler beams. Our setup has beam waists of $w_p = 26\,\mu\mathrm{m}$ for the pump and $w_s = w_i = 36\,\mu\mathrm{m}$ for the signal and idler, located at position $z_{0,j}$ inside the crystal for $j \in \{p, s, i\}$.

The probability of photon-pair collection is given as

$$\tau(t) \propto \begin{cases} 0 & |t| > \frac{L}{2v_g} \\ \mathcal{O}_t(t)^2 & \text{else} \end{cases},$$ (A10)

with crystal length $L$ and group velocity $v_g$. Values for $n = n_y(403.9\,\mathrm{nm}) = 1.841$ at a temperature of $T = 31.9\,°\mathrm{C}$, $v_o = c/1.805$ and $v_e = c/1.910$ are taken from [35] and [37] but are also in good agreement with [36].

Convolving equation (A9) with equation (A10) and renormalizing gives a temporal distribution of the photon arrival times at the PBS of the Sagnac interferometer for the ↻ and ↺ direction $Q_j(t) \equiv \tau_j(t) \circledast I_p(t)$, where the ↻ and ↺ distributions are different in their sign of $z_0$.

To estimate the influence of an off-centered crystal on the fidelity, concurrence and Bell-CHSH $S$ parameter, we numerically calculated $\tau_o(t)$ and $\tau_e(t)$ for both ↻ and ↺ directions then, convolved them with $I_p(t)$ and finally calculated the overlap integral $\mathcal{O}_c^j$ between the ↻ and ↺ directions for both polarizations ($j \in \{o, e\}$) present in the type-II down-conversion process, such that with crystal position $z_c$

$$\mathcal{O}_c^j(z_c) = \left( \int dt \sqrt{Q_j(t)\big|_{z_0 = z_c} \cdot Q_j(t)\big|_{z_0 = -z_c}} \right)^2,$$ (A11)

and $z_0 = \pm z_c$ corresponds to ↻ and ↺ directions, respectively. The resulting state, after interference given a crystal position $z_c$ away from the Sagnac interferometer's center is then defined by

$$\rho(z_c) = \frac{1}{2}(\mathcal{O}_c^o + \mathcal{O}_c^e)|\Psi^-\rangle\langle\Psi^-| +$$ (A12)
$$\frac{2 - \mathcal{O}_c^o - \mathcal{O}_c^e}{4}(|HV\rangle\langle HV| + |VH\rangle\langle VH|).$$

From this density matrix, we calculate the expected CHSH $S$ parameter and plot $2\sqrt{2} - S$ in Fig. 5b in the main text.

It is clear that the position of the collection foci can reduce the $S$ parameter as the birefringent walk-off is no longer compensated by the HWP ($\mathcal{O}_c^o + \mathcal{O}_c^e < 2$). For a maximally entangled Bell-state, the focal point for signal and idler must be at the center of the nonlinear crystal. We also compare our simulation to previous results using the same Sagnac interferometer. In Fig. A7, we plot the simulated concurrence as well as experimental results [13,38]. A similar trend is visible when we offset the theoretical curve to match the maximal concurrence measured in the experiment.

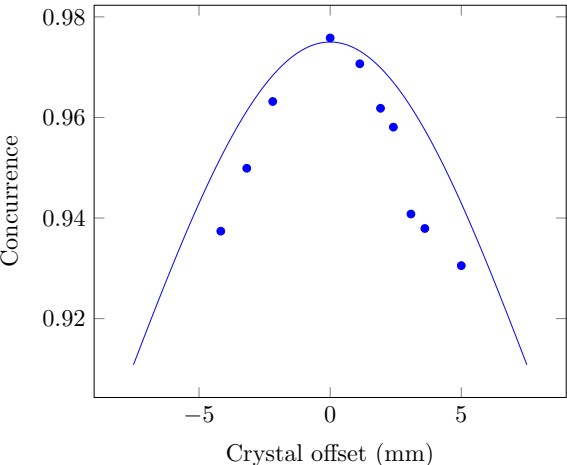

Figure A7: Quantum state concurrence reduces as the crystal is offset from the center of the Sagnac interferometer. The theoretical curve (solid) was scaled such that its maximum coincides with the concurrence of 0.982 reported in [13,38].

# F  Measure of multi-pair emission

For the $\circlearrowright$ emission direction, the single photon count rate in output mode A(B) of the Sagnac interferometer is given as $R_A = \nu\eta_A$ ($R_B = \nu\eta_B$) where $\nu$ is the pair emission rate rate in this direction and $\eta_{A(B)}$ is the channel transmission, including the polarizers and single photon detector efficiency. The coincidence rate is then given as $R_c = \nu\eta_A\eta_B$ and the emission rate as $\nu = \frac{R_A R_B}{R_c}$. With the same pump power as in the main experiment and the polarizers set to $H$ and $V$, we measure an average single count for channel A and B as $17\,380\,\text{s}^{-1}$, $17\,458\,\text{s}^{-1}$ respectively and the coincidence rate as $2181\,\text{s}^{-1}$. This gives $\nu = 139\,\text{kHz}$ and collection efficiencies of $\eta_A \approx \eta_B \approx 0.125$. Due to the high degree of symmetry between the emission directions, the total generated pair rate is $R_\text{pair} \approx 2\nu = 278\,\text{kHz}$.

As the coherence times of the down-converted photons as well as of the pump laser (typ. 3 ps) are both much shorter than the coincidence window $T_c = 96\,\text{ps}$, multi-pair detection is dominated by independent spontaneous emissions (accidentals) occurring within $T_c$. The rate of these emissions is then given by $T_c R_\text{pair}^2/2$, which yields a ratio of multi-pairs to pairs as $p = \frac{T_c R_\text{pair}^2}{2R_\text{pair}} = \nu T_c$, i.e. $p$ equals the probability to generate a single pair in a single direction within $T_c$. For our experiment, we obtain $p = 1.34 \times 10^{-5}$. A coincidence measurement arising from multi-pair emission can be triggered from all combinations of emission directions (double-emission in one direction or balanced emission in both directions). Considering these combinations as well as the fact that the detectors are not photon-number resolving, one can

derive the rate of twofold coincidences from double-pair emissions ($R_4$) relative to the measured coincidences as a function of $p$ and the polarizer angles $\alpha$ and $\beta$:

$$\frac{R_4(\alpha, \beta, p)}{R_c} = \frac{p}{2} \Big[ \left(2 - \eta_A \sin^2(\alpha)\right) \left(2 - \eta_B \cos^2(\beta)\right) \left(\sin(\alpha)\cos(\beta)\right)^2 +$$
$$\left(2 - \eta_A \cos^2(\alpha)\right) \left(2 - \eta_B \sin^2(\beta)\right) \left(\cos(\alpha)\sin(\beta)\right)^2 +$$
$$2 \left(1 - \eta_A(\sin(\alpha)\cos(\alpha))^2\right) \left(1 - \eta_B(\sin(\beta)\cos(\beta))^2\right) \Big]. \qquad \text{(A13)}$$

We can now estimate the impact on the Bell-CHSH experiment from Eq. (A13) by considering the count rates for the 16 projection measurements and find the resulting value of $2\sqrt{2} - S = 1.09 \times 10^{-4}$. We estimate the impact of the multi-pairs on fidelity and concurrence of the reconstructed two-photon quantum state by calculating the expected count rate for each projection measurement and performing state tomography with these counts (without MLE). The results are displayed in Fig. 5c of the main paper.

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
