# Peer review of "Approaching the Tsirelson bound with a Sagnac source of polarization-entangled photons"

_SciPost Physics, doi:SciPost Phys. 10, 017 (2021)_

## Round 1 · Referee Report · Anonymous (Referee 1) · 2020-12-10

Report

This is an interesting paper that reports the realization of a Sagnac source of polarization-entangled photons that achieves a very high value of the Bell-CHSH S parameter, while also achieving a high rate. The authors analyze different sources of imperfection and describe a path towards further improvements of the S parameter, which might allow them to break the current record (to which they are already quite close).

I think that this paper may well deserve publication in SciPost, but I do have one main question. The authors state that the main source of error is the imprecision in the location of the focal point, which they estimate to be about 1 mm. I see no reason to doubt their analysis that this level of imprecision would roughly explain the reduction in S that they observe. But I didn't see a clear demonstration that this is in fact what limits their experiment. The authors don't give a lot of detail on how they obtained their estimate of 1 mm. Would it maybe be more accurate to say that they infer this level of imprecision from the fact that the other considered reasons can't explain the observed reduction in S value? The authors don't seem to think so, since they state that there is still a portion of the reduction in S that is unexplained. But how sure are they really that their imprecision is exactly 1 mm? Could it be even more? On the other hand, 1 mm does seem relatively large. How hard would it be to try to reduce this value in the experiment? If the authors are right, then this should increase S and potentially bring them into record-breaking territory. The authors say that the error sources that they have identified here can be readily improved, but I assume that if it had been completely straightforward to do, they would have implemented especially the most relevant improvement (fixing the focal point location) already in this experiment. I think that the authors should at least discuss these points more clearly.
  • validity: -
  • significance: -
  • originality: -
  • clarity: -
  • formatting: -
  • grammar: -

Author:  Robert J. Chapman  on 2020-12-18  [id 1090]

(in reply to Report 1 on 2020-12-10)
Category:
answer to question

We thank the referee for reviewing our work and deeming it of interest for publication in SciPost.

We determine the 1 mm precision by the fact that we use a ruler with 1 mm divisions to position the crystal at the centre of the Sagnac loop and by the fact that the ruler cannot be placed in the actual beam path. A 1 mm offset of the focal position accounts for almost half of the difference between our measured $S$ parameter and the Tsirelson bound. We have added the following sentence to the manuscript:
"We estimate $1.0$ mm accuracy of our ppKTP crystal position from the center of the Sagnac interferometer by the precision of the ruler used to measure it."

Indeed, as the referee suggests, we consider several other error sources that contribute to the reduction in the $S$ parameter in the manuscript. Ultimately, we consider 1 mm a reasonable level of imprecision for the focal position, and this contributes the largest decrease in the S parameter.

As the referee proposes, we could also infer the focal offset from the gap to the $S$ parameter, which would correspond to 1.5 mm from the centre of the Sagnac loop. It is plausible that this explains our gap to the S parameter; however, we would avoid using this as an argument for determining the focal offset. We have added the following statement to the conclusion of the manuscript:
"A $1.5$ mm offset of the focal positions inside the crystal would account fully for this gap, however, at this point we cannot rule out further error sources."

Performing state tomography or the CHSH experiment for each position of the focal point inside the crystal would enable us to achieve a higher level of precision and, as the components are on translation stages, we could improve the precision by an order of magnitude. This is a laborious approach, as the measurement at each steps take significant time, however, this would be necessary to close the gap to the Tsirelson bound. Indeed, in a previous work we have measured the concurrence of the state while moving the nonlinear crystal and have demonstrated what we predict to follow from the theory: see references 13 and 38 in the manuscript and Supplementary Section E and Figure A7.

---

## Round 1 · Referee Report · Anonymous (Referee 2) · 2020-12-14

Report

This manuscripts presents an experiment to approach Tsirelson's bound using a Sagnac photon source generating polarization entangled photons. The authors come by $5.65 \cdot 10^{-3}$ close to Tsirelson's bound with a high rate of detected photons of 4660 pairs/s/mW. Interestingly, the authors provide an analysis of the origin of the deviation of Tsirelson's bound by identifying particular aspects of their experimental implementation, which can serve as a guide for future work in this field.

This work is well written, clear and of interest for the community in reaching for Tsirelson's bound using photon pairs. I can therefore recommend publication in SciPost Physics, once the authors provide some clarifications and fixing of typos as outlined in the following:

1) The authors focus very much on identifying the experimental reasons of approaching Tsirelson's bound. However, the manuscript also emphasizes that the experiment generates photons with a high rate. The origin of the high rate compared to previous experiments is, unfortunately, not explained. Is this solely due to using superconducting single photon detectors or using a longer crystal than in previous experiments? The authors are kindly asked to comment on this.

2) The authors write on page 4 that "A bipartite quantum state with S > 2 cannot be described by local theories...". Shouldn't it be rather: "A bipartite quantum state with S > 2 cannot be described by local-realistic theories..." as the Bell-CHSH inequality makes the assumption that local-realism holds.

3) Typo: page 5 top: detailled -> detailed

4) Typo: Eq. 7: A "(" too much in the first line. This is also the case for eq. A12

5) Page 8 bottom: The authors write: "There is still a gap to the Tsirelson bound of...". I would rather suggest: "There is still a gap between the experimentally recorded value and the predicted value of ..."

6) Appendix D, Figure A4b: The concurrence in the figure is about 0.91. However, in the text the authors mention the value of the concurrence as 0.965. Some value is wrong here.

7) Please clarify the notation after eq. A9 of $O_t=O_s(tv_g)$. Though $O_s$ itself is defined in the main text, it now receives an argument of $tv_g$, where it is not clear how this argument translates to the variables used in eq. 5. Further, it does not become clear, why the authors need to introduce the definition for $O_t$ as they could use $O_s$ in the following as well. This seems unnecessary, but maybe I have overlooked something.

8) Typo: In the paragraph right before eq. A11 it should be $I_p(t)$ and not $I(t)$.

Requested changes

Copied from the report above:

1) The authors focus very much on identifying the experimental reasons of approaching Tsirelson's bound. However, the manuscript also emphasizes that the experiment generates photons with a high rate. The origin of the high rate compared to previous experiments is, unfortunately, not explained. Is this solely due to using superconducting single photon detectors or using a longer crystal than in previous experiments? The authors are kindly asked to comment on this.

2) The authors write on page 4 that "A bipartite quantum state with S > 2 cannot be described by local theories...". Shouldn't it be rather: "A bipartite quantum state with S > 2 cannot be described by local-realistic theories..." as the Bell-CHSH inequality makes the assumption that local-realism holds.

3) Typo: page 5 top: detailled -> detailed

4) Typo: Eq. 7: A "(" too much in the first line. This is also the case for eq. A12

5) Page 8 bottom: The authors write: "There is still a gap to the Tsirelson bound of...". I would rather suggest: "There is still a gap between the experimentally recorded value and the predicted value of ..."

6) Appendix D, Figure A4b: The concurrence in the figure is about 0.91. However, in the text the authors mention the value of the concurrence as 0.965. Some value is wrong here.

7) Please clarify the notation after eq. A9 of $O_t=O_s(tv_g)$. Though $O_s$ itself is defined in the main text, it now receives an argument of $tv_g$, where it is not clear how this argument translates to the variables used in eq. 5. Further, it does not become clear, why the authors need to introduce the definition for $O_t$ as they could use $O_s$ in the following as well. This seems unnecessary, but maybe I have overlooked something.

8) Typo: In the paragraph right before eq. A11 it should be $I_p(t)$ and not $I(t)$.

  • validity: high
  • significance: good
  • originality: good
  • clarity: high
  • formatting: perfect
  • grammar: perfect

Author:  Robert J. Chapman  on 2020-12-18  [id 1091]

(in reply to Report 2 on 2020-12-14)
Category:
answer to question

We thank the referee for their thorough review of our manuscript and recommending it for publication in SciPost Physics. Please see below our response to the referees comments.

  1. The high brightness we achieve is due to co-linear photon pair generation, meaning a large fraction of the generated photons can be collected in single mode fibers. In contrast, cone sources discard the majority of photon pairs (as only a section of the cone can be probed by fibers), meaning the source brightness in pairs/second per milliwatts of pump power is much lower. Indeed our experiment does benefit from SNSPDs with around 60% efficiency; however, the record-violation CHSH experiments utilize silicon APDs operating around 800 nm wavelength with 40% quantum efficiency, but have nearly two orders-of-magnitude lower brightness. We have added the following statement to the manuscript on page 4: “We achieve high brightness by constructing a well-aligned colinear photon-pair source, where all generated photons can be utilized, and by optimizing the output collection into single mode optical fibers. Cone sources suffer from low brightness as most of the photons are discarded, meaning far higher pump powers are required for the same fiber-coupled photon-pair flux. We measure around two orders of magnitude higher brightness than the cone source with the record high $S$ parameter [23].”
  2. We thank the referee for pointing out this detail; we have included their suggestion in the manuscript.
  3. Typo corrected in manuscript
  4. Typo corrected in manuscript
  5. We thank the referee for their suggestion to make this statement clearer. We have updated the manuscript.
  6. This is a typo in the text, the correct value is 0.910\pm0.005
  7. We have updated the manuscript to read $\mathcal{O}_s^j(t) = \mathcal{O}_t(v_g^jt)$ with $j \in {s,i,p}$.
  8. Typo corrected in manuscript

---

## Round 2 · Referee Report · Anonymous (Referee 2) · 2021-1-13

Report

The authors have addressed the comments I made in a satisfactory way and added clarifications to the manuscript. Thanks a lot!

---

## Round 2 · Referee Report · Anonymous (Referee 1) · 2021-1-14

Report

I think the authors have adequately responded to my comments.

---

## Round 2 · Author Response

Changes implemented as per the referees' comments.

---

## Editorial Decision

published